# Life without Proteinase Activated Receptor 2 (PAR2) Alters Body Composition and Glucose Tolerance in Mice

**DOI:** 10.3390/nu14194096

**Published:** 2022-10-02

**Authors:** Thomas H. Reynolds, Stephen J. Ives

**Affiliations:** Health and Human Physiological Sciences, Skidmore College, Saratoga Springs, NY 12866, USA

**Keywords:** obesity, diet, aging, nutrient sensing, insulin action, glucoregulation, AMPK

## Abstract

The potential role of proteinase activated receptor 2 (PAR2) in the development of age-related obesity and insulin resistance is not well-understood. To address the hypothesis that deletion of PAR2 might ameliorate age-related obesity and impaired glucose homeostasis, we assessed body composition and insulin action in 18-month-old male PAR2 knockout (PAR2KO-AG), age-matched (AG) and young C57BL6 (YG, 6-month-old) mice. Body composition was measured by magnetic resonance spectroscopy (MRS) and insulin action was assessed by glucose tolerance (GT), insulin tolerance (IT) and AICAR tolerance (AT) testing. AG mice weighed significantly more than YG mice (*p* = 0.0001) demonstrating age-related obesity. However, PAR2KO-AG mice weighed significantly more than AG mice (*p* = 0.042), indicating that PAR2 may prevent a portion of age-related obesity. PAR2KO-AG and AG mice had greater fat mass and body fat percentage than YG mice. Similar to body weight, fat mass was greater in PAR2KO-AG mice compared to AG mice (*p* = 0.045); however, only a trend for greater body fat percentage in PAR2KO-AG compared to AG mice was observed (*p* = 0.09). No differences existed in lean body mass among the PAR2KO-AG, AG, and YG mice (*p* = 0.58). With regard to insulin action, the area under the curve (AUC) for GT was lower in PAR2KO-AG compared to AG mice (*p* = 0.0003) and YG mice (*p* = 0.001); however, no differences existed for the AUC for IT or AT. Our findings indicate that age-related obesity is not dependent on PAR2 expression.

## 1. Introduction

Obesity is a major public health problem in the United States [1,2], accounting for almost 8% of healthcare expenditures or USD 342.2 billion for non-institutionalized adults [3]. Worldwide the prevalence of overweight and obesity is estimated to be 30% of the world’s population, a figure that is expected to increase to 50% in less than a decade [4]. Further, aging appears to impact obesity; in the United States, the age-adjusted prevalence of severe obesity is greater in adults ages 40–59 than adults ages 20–39 [2]. This is particularly concerning since the number of individuals between the ages of 45–64 and over 65 year of ages is expected to increase by 19.8% and 112%, respectively, by 2060 [5]. Despite the increased prevalence of obesity in the United States and worldwide, there are limited treatment options to combat this devastating disease.

Numerous pre-clinical studies in rodents have been conducted in the hopes of finding an effective treatment for obesity. According to the National Library of Medicine’s PubMed.gov search portal, over the last ten years more than 10,000 studies have examined a high-fat diet in wild type and genetically altered mice (https://pubmed.ncbi.nlm.nih.gov, 3 June 2021). These studies have certainly increased our knowledge regarding factors that regulate adiposity and the associated negative impact on insulin action; however, the rapid weight gain typically observed in mice fed a high-fat or Western diet may not be an ideal model for the slow insidious weight gain observed in humans as they approach mid-life. Along these lines, a recent pre-clinical study in mice indicates that interventions that reverse diet-induced obesity may not affect age-related obesity [6]. Thus, perhaps age-related obesity is a better pre-clinical obesity model for human obesity than high-fat feeding regimens, but more studies are needed to gain a better understanding of the cellular mechanisms that govern age-related obesity.

Proteinase activated receptor 2 (PAR2) is a G-protein coupled receptor that plays a role in inflammation and has been implicated in the development of diet-induced obesity [7,8,9]. PAR2 is activated by cleavage of its N-terminal domain by tissue factor and other serine proteases [10], a process that can lead to reduced AMPK activity [11,12]. Recently, pro-inflammatory cytokines and insulin resistance were associated with hepatic expression of PAR2 aged rats, while in HepG2 cells PAR2 deletions appears to reduce insulin resistance and inflammation; however, in livers from PAR2 knockout mice IRS/Akt signaling was increased [13]. Although a PAR2 antagonists can prevent diet-induced obesity and insulin resistance [8], the role PAR2 plays in the development of aged-related obesity and insulin resistance is not currently known.

To date, not studies have examined the impact of PAR2 expression on age-related obesity and insulin resistance. Therefore, the purpose of the present study is to determine if life without PAR2 prevents the development of obesity and insulin resistance that occurs with advancing age. To accomplish this, we assessed body composition and insulin action in young (YG), aged (AG), and aged PAR2 knockout (PAR2KO-AG) mice. We hypothesized that deletion of PAR2 would abolish, or attenuate, the age-associated obesity and accompanying impairment in insulin action.

## 2. Materials and Methods

### 2.1. Animals

Male PAR2 knockout mice (n = 22, Stock #004993) and C57BL6 (n = 29, Stock #000664) were purchased from Jackson Laboratories (Jackson, Bar Harbor, ME, USA) at the age of eight weeks and were housed until 18 months of age. Approximately 12 months later, an additional group of male C57BL6 mice (n = 8) were purchased from Jackson Laboratories at eight weeks of age and housed until six months of age. All mice were fed Prolab RMH 3000 normal chow (Lab Diet, St. Louis, MO, USA). The mice were on a 12:12 h light:dark cycle with the lights turning on at 8:00 AM and off at 8:00 PM. The temperature and humidity of the animal housing room was approximately 23 °C and 40%, respectively. Figure 1 provides a schematic overview of the study design. The current study followed the guidelines set forth by the National Research Council’s Guide for Care and Use of Laboratory Animals (Institute of Laboratory Animal Resources, Commission on Life Sciences, 2011), and the experimental protocol was approved by the Skidmore College Institutional Animal Care and use Committee (Protocol #174, Approved on 25 July 2019).

### 2.2. Glucose Tolerance Testing

For glucose tolerance (GT) testing, mice were randomly selected from each group so that a total of 20 mice were studied (YG, n = 7; AG, n = 6; and PAR2KO-AG, n = 7). The randomly selected mice received an intraperitoneal injection of glucose (2.0 g/kg body weight) following an overnight fast (~15 h). Tail vein blood was collected (3–5 μL) at 0, 20, 40, 60, and 90 min following the injection and blood glucose was measured using a hand-held glucometer (Accu-Check, Roche Diabetes Care, Inc., Boston, MA, USA) in duplicate, and averaged. Mice were allowed to recover for 7–10 days following the GT testing before they were subjected to insulin tolerance testing.

### 2.3. Insulin Tolerance Testing

Because it is difficult to obtain sufficient blood volume to assess insulin during a GT, we conducted insulin tolerance (IT) testing to gain insights regarding insulin sensitivity. For IT tests, the same mice used for GT testing were injected with insulin (0.75 U/kg body weight) following a 6 h fast. Tail vein blood was collected (3–5 μL) at 0, 20, 40, and 60 min following the injection and blood glucose was measured using a hand-held glucometer (Accu-Check, Roche Diabetes Care, Inc.) in duplicate, and averaged. Mice were allowed to recover for 7–10 days following the IT testing before they were subjected to *N*^1^-(β-D-Ribofuranosyl)-5-aminoimidazole-4-carboxamide (AICAR) tolerance testing.

### 2.4. AICAR Tolerance Testing

To gain further information concerning AMPK-stimulated glucose metabolism, we conducted a tolerance test using *N*^1^-(β-D-Ribofuranosyl)-5-aminoimidazole-4-carboxamide (AICAR). For AICAR tolerance (AT) testing, the same mice used for the GT and IT testing were used and the procedure was identical to IT testing except mice were injected with AICAR (250 mg/kg). Tail vein blood was collected (3–5 μL) at 0, 20, 40, and 60 min following the injection and blood glucose was measured using a hand-held glucometer (Accu-Check, Roche Diabetes Care, Inc.) in duplicate, and averaged.

### 2.5. Body Composition

An LF50-BCA Minispec (Bruker Inc., Bilierica, MA, USA) assessed body composition in mice. The Minispec is a nuclear magnetic resonance (NMR) system that allows for the quantification of fat mass, lean mass, and free body fluid. NMR scans were completed by immobilizing fully conscious mice in a plastic tube that was placed in the instrument’s sample chamber for approximately 90 s. Body composition in mice assessed by NMR is highly correlated to the chemical analysis method of body composition assessment [14]. Further, epididymal adipose tissue (EAT) mass was assessed in YG mice (n = 8) and a subset of AG (n = 6) and PAR2KO-AG (n = 8) mice. Though we have previously documented that EAT and NMR are highly related [15]. The remaining AG and PAR2KO-AG mice were not euthanized and used for a separate study.

### 2.6. Statistical Analysis

All statistical analyses were carried out using commercially available software (StatView, SAS Institute, Cary, NC, USA). Two-way mixed analysis of variance (ANOVA) was used to compare the glucose response to the in vivo tolerance tests of insulin action between groups over time. A one-way ANOVA was used to detect statistical differences in the area under the curve (AUC) of the in vivo assessments of insulin action, body weight and composition parameters between PAR2KO-AG, AG, and YG mice. An HSD post hoc analysis was used to locate the significance. Data are presented as mean ± SD, and the level of statistical significance was set at *p* < 0.05.

## 3. Results

### 3.1. GT Testing

To gain initial insights into the role of aging and PAR2 on insulin action in we conducted intraperitoneal GT testing in YG, AG, and PAR2KO-AG mice. At baseline, fasting glucose levels were significantly higher in AG and PAR2KO-AG mice compared to YG mice; however, no differences existed between AG and PAR2KO-AG mice. As shown in Figure 2, glucose values during the GT test were significantly higher in YG and AG mice compared to PAR2KO-AG mice. These findings indicate that PAR2 may play a role in either insulin sensitivity or insulin secretion.

### 3.2. IT Testing

To determine if the differences in GT were due to changes in insulin sensitivity, we conducted IT test. The IT test involves an intraperitoneal injection of insulin at a dose that would be expected to suppress insulin secretion and allow for an evaluation of peripheral insulin sensitivity. As shown in Figure 3, no differences were observed in IT among YG, AG, and PAR2KO-AG mice. These findings indicate that the greater GT in PAR2KO-AG mice were likely not related to changes in insulin sensitivity but rather insulin secretion.

### 3.3. AT Testing

Since insulin action was similar in YG, AG, and PAR2KO-AG mice, we tested if glucose metabolism was different following activation of AMPK by an intraperitoneal injection of AICAR. As shown in Figure 4, AICAR reduced blood glucose values in all mice; however, there were no significant differences among YG, AG, and PAR2KO-AG mice. These findings indicate that the non-insulin dependent stimulation of glucose metabolism is not altered by aging or the life-long absence of PAR2.

### 3.4. Body Composition, Body Mass, and EAT Mass

To determine if PAR2 plays a role in the development of age-related obesity, we assessed body mass, EAT mass, and body composition in YG, AG, and PAR2KO-AG mice (Table 1). As expected, aging resulted in a significant increase in body mass and this age-related increase was exacerbated in PAR2KO-AG mice. The increase in body mass with age and PAR2 deletion appears to be due to an increase in fat mass. Fat mass was significantly higher in AG compared to YG mice; further, PAR2KO-AG mice had significantly more fat mass than AG. However, in a small subset (n = 6–8), no differences existed in EAT mass between AG and PAR2KO-AG mice, although significantly greater than EAT mass for YG mice. No differences existed in lean body mass across the YG, AG, and PAR2KO-AG mice.

## 4. Discussion

The purpose of the present study was to determine if PAR2 plays a critical role in the development of age-related obesity and insulin resistance. To accomplish this, we assessed body composition and insulin action in young, aged, and aged PAR2 knockout mice. The most interesting observation was that life without PAR2 did not attenuate the increase in adiposity with advancing age. In fact, the PAR2KO-AG mice possessed greater body mass and fat mass than C57BL6 mice of the same age. Concerning insulin action, despite greater glucose tolerance, PAR2KO-AG mice have similar insulin and AICAR tolerance when compared to YG and AG mice, suggesting enhanced insulin secretion over altered sensitivity. Collectively, these findings suggest that PAR2 does not attenuate age-associated obesity or insulin action. This observation is uniquely different from studies that have implicated PAR2 in diet-induced obesity and insulin resistance in young wildtype and PAR2KO mice [8,9]. Despite our interesting findings, our results are limited by studying mice at a housing temperature below typical threshold for thermoneutrality and therefore all mice expended energy to maintain body temperature. Our results are also limited by assessing insulin sensitivity by IT testing, rather than by a hyperinsulinemic-euglycemic clamp or serial insulin measurements during GT testing. Finally, we only assessed body weight, body composition, and glucose metabolism at the end of our study, so we cannot rule out temporal changes that may have occurred earlier in the aging process.

It is well-established that tissue factor-PAR2 signaling plays a role in the development of diet-induced obesity [8,9,12]. Badeanlou et al. [9] demonstrated that tissue factor and PAR2 regulate body weight in 19–26-week-old male C56BL/6 mice and that adipose tissue mass was lower in PAR2KO mice following 16 weeks of a high-fat diet. Further, plasma tissue factor and visceral adiposity tissue mRNA were higher in fat-fed mice and treating mice with a highly specific PAR2 inhibitor prevented diet-induced obesity [8,9]. However, in our hands PAR2KO mice were not protected from age-related obesity when studied at 18 months of age. In fact, PAR2KO-AG actually had greater fat mass than aged-matched C57BL6 mice, a surprising finding running counter to the idea that PAR2 plays a critical in the development of obesity (Table 1). However, EAT mass was not different between AG and PAR2KO-AG mice, suggesting that visceral fat depots do not contribute to the increased total fat mass assessed by MRS. Because EAT mass was assessed in only a subset of AG and PAR2KO-AG mice (n = 6–8) due to being part of another in vivo study, this interpretation should be taken with caution. Perhaps if we had evaluated body composition at younger age (5–7 months old) as Badeanlou et al. [9] did, we may have implicated PAR2 as a regulator of adiposity; however, the primary goal of our study was to examine the role of PAR2 on age-related obesity.

It is quite possible that aging with a complete absence of PAR2 leads to a compensatory adaptation that changes the phenotype observed in young mice. Interestingly, Maruyama et al. [16] demonstrated an age-related increase in body mass with a decline in PAR2 expression in aortas of spontaneous hypertensive rats (SHRSO.ZF), suggesting, albeit not in adipose tissue, that PAR2 expression is not related to weight gain. Further it is important to point out that interventions that prevent high-fat diet induced obesity may, or may not, reduce age-related obesity [6,17], indicating that the slow gradual increase in adiposity with aging is distinct from the rapid weight gain observed with high-fat feeding regimens.

In addition to a role in diet-induced obesity, PAR2 signaling has also been implicated in the development of insulin resistance. Inhibiting PAR2 signaling with either a highly specific PAR2 inhibitor or a tissue factor antibody appears to improve both glucose tolerance and insulin tolerance in obese rats and mice [8,9]. Interestingly, Badeanlou et al. [9] observed a significant improvement in glucose and inulin tolerance in obese mice 24 h following an injection of a rat antibody to mouse tissue factor, indicating that blocking a major endogenous activator of PAR2 signaling improved insulin action independent of weight loss. Whereas the PAR2 antagonist mediated improvement in insulin action observed by Lim et al. [8] was associated with a reduction in adiposity. Recently, inhibiting neutrophil elastase, a PAR2 activating serine proteinase improved insulin sensitivity in adipose tissue [18]. Likewise, Kim et al. [13] observed an increase in insulin signaling in liver tissue from young PAR2KO mice as well as in HepG2 cells transfected with PAR2-siRNA. Perhaps PAR2 plays a more prominent role in hepatic insulin action compared to peripheral tissues (skeletal muscle). In the present study, we observed no effect of aging with or without PAR2 on insulin sensitivity as assessed by an IT test. This observation is not surprising as we and others have previously shown that aging does not result in insulin resistance in male C57BL6 mice [6,19,20], although Oh et al. report insulin resistance in aged mice [21]. However, area under the glucose tolerance curve was significantly reduced in PAR2KO-AG mice, suggesting that PAR2 may restrain insulin secretion in vivo in a similar fashion as it does in cultured MIN6 β cells [8]. Further, PAR2 may play a more prominent role in hepatic insulin action compared to peripheral tissues (skeletal muscle) as Kim et al. [13] observed an increase in insulin signaling in liver tissue from young PAR2KO mice as well as in HepG2 cells transfected with PAR2-siRNA.

Further, muscle contraction is also well-established to increase glucose transport into skeletal muscle, a phenomenon that is believed to be regulated by AMP activated protein kinase (AMPK) [22]. Although in some cell types PAR2 can activate AMPK, evidence indicates that AMPK signaling is impaired by PAR2 signaling [12] in cells that express high levels of the scaffold protein β-arrestin [23]. Interestingly, obesity and type 2 diabetes are associated with impaired AMPK activity [24,25] and metformin, a widely prescribed drug to treat hyperglycemia, may lower blood glucose by activating AMPK [26]. To probe a possible connection between PAR2 signaling and AMPK we assessed blood glucose levels in response to the AMPK agonist AICAR in aged PAR2 knockout mice and aged-matched C57BL/6 mice. Although AICAR reduced blood glucose, this effect appears to be independent of PAR2.

## 5. Conclusions

Contrary to our hypothesis, genetic ablation of PAR2 did not attenuate the increase in adiposity associated with advanced age; the PAR2KO-AG mice actually exhibited greater body mass and fat mass than control C57BL6 mice of the same age. This is an interesting finding that is distinct from the role of PAR2 and diet-induced obesity in young mice. Regarding insulin action, despite greater glucose tolerance, PAR2KO-AG mice have similar insulin and AICAR tolerance when compared to YG and AG mice, suggesting that knockout of PAR2 altered insulin secretion and/or clearance but had no effect on insulin sensitivity. These present findings suggest that PAR2 does not play an important role in the development of age-associated obesity and insulin resistance. Perhaps future studies can examine the effect of PAR2 antagonists on age-related obesity.

## Figures and Tables

**Figure 1 nutrients-14-04096-f001:**
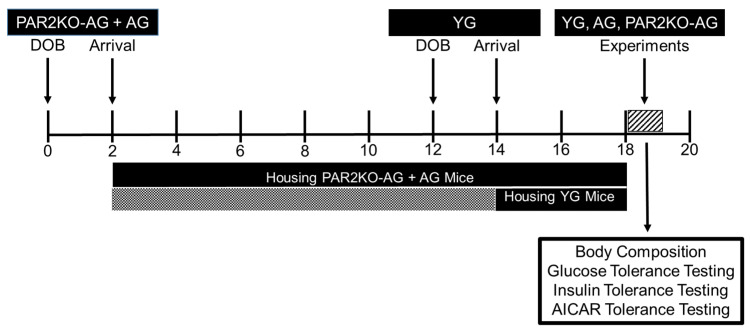
Overview of the experimental design.

**Figure 2 nutrients-14-04096-f002:**
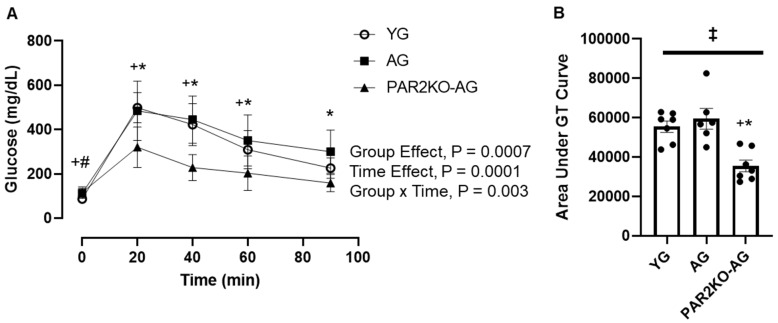
Glucose tolerance in YG (n = 7), AG (n = 6), and PARKO-AG (n = 7) male mice. Mice received an intraperitoneal injection of glucose (2 mg/kg body weight), and blood glucose was assessed at indicated time points following the injection (**A**). The glucose tolerance test area under the curve (AUC) was calculated from blood glucose values obtained at baseline and following glucose injection (**B**). Data were analyzed using a repeated measure (**A**) and one-way (**B**) ANOVA. * PAR2KO mice significantly different from AG mice, *p* ≤ 0.05, ^+^ PAR2KO mice significantly different from YG mice, *p* ≤ 0.05, ^#^ YG mice significantly different from AG mice, *p* ≤ 0.05, ^‡^ Significant 1 × 3 ANOVA, *p* ≤ 0.05.

**Figure 3 nutrients-14-04096-f003:**
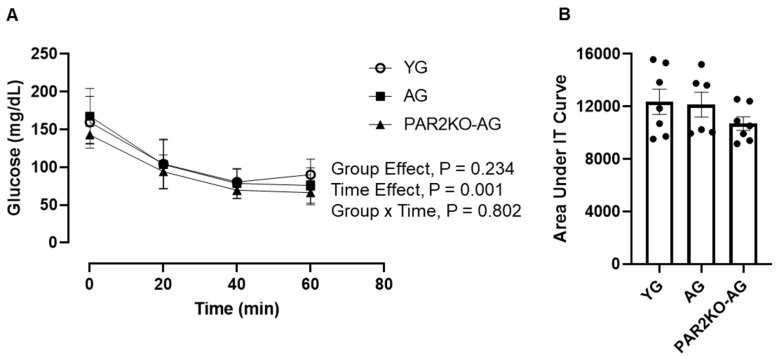
Insulin tolerance in YG (n = 7), AG (n = 6), and PARKO-AG (n = 7) male mice. Mice received an intraperitoneal injection of insulin (0.75 U/kg body weight), and blood glucose was assessed at indicated time points following the injection (**A**). The insulin tolerance test area under the curve (AUC) was calculated from blood glucose values obtained at baseline and following insulin injection (**B**). Data were analyzed using a repeated measures (**A**) and one-way (**B**) ANOVA.

**Figure 4 nutrients-14-04096-f004:**
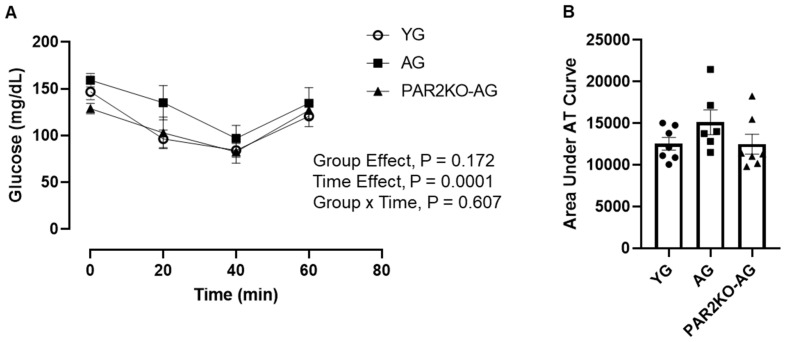
AICAR tolerance in YG (n = 7), AG (n = 6), and PARKO-AG (n = 7) male mice. Mice received an intraperitoneal injection of AICAR (250 mg/kg body weight), and blood glucose was assessed at indicated time points following the injection (**A**). The AICR tolerance test area under the curve (AUC) was calculated from blood glucose values obtained at baseline and following AICAR injection (**B**). Data were analyzed using a repeated measures (**A**) and one-way (**B**) ANOVA.

**Table 1 nutrients-14-04096-t001:** Body composition and epididymal white adipose tissue (EAT) of young, aged, and aged PAR2 knockout mice.

Parameter	YG (n = 8)	AG (n = 29)	PAR2KO-AG (n = 22)	ANOVA *p*-Value
Body Mass (g)	28.88 ± 0.75	44.20 ± 1.1 ^#^	47.79 ± 1.5 *	0.0001
Fat Mass (g)	6.15 ± 0.63	19.42 ± 1.43 ^#^	23.62 ± 1.65 *	0.0001
Lean Mass (g)	19.04 ± 0.54	19.1 ± 0.48	18.86 ± 0.42	0.584
Body Fat (%)	21.16 ± 1.95	42.55 ± 2.42 ^#^	48.15 ± 2.23 ^#,^^┬^	0.0001
EAT Mass (g) ^¥^	0.38 ± 0.05	1.22 ± 0.07 ^#^	1.21 ± 0.09 ^#^	0.0001

^¥^ The number of mice for EAT mass was 6–8. * Denotes a significant difference from AG and YG mice, *p* ≤ 0.05. ^#^ Denotes a significant difference from YG mice, *p* ≤ 0.05. ^┬^ Denotes a trend for a difference from AG mice, *p* = 0.088.

## Data Availability

The data are available upon reasonable request.

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
