# Peer review of "Life without Proteinase Activated Receptor 2 (PAR2) Alters Body Composition and Glucose Tolerance in Mice"

_nutrients, 2022, doi:10.3390/nu14194096_

Round 1
Reviewer 1 Report
Title: Life Without Proteinase Activated Receptor 2 (PAR2) Alters 2 Body Composition and Glucose Tolerance in Mice
Although the study looks interesting there are major issues with this manuscript.
English language throughout the manuscript needs to be improved
ABSTRACT
Comment 1: Results section need to be rewritten clearly, it’s a bit confusing to the reader.
INTRODUCTION
Comment 2: The authors laid a proper foundation, identified the gap in knowledge, clearly outlined the research questions being addressed by the paper and provided justification for doing the study, but novelty of the studies yet to be cleared.
MATERIALS AND METHODS
Comment 3: The author should provide animal ethics committee details (Registration details of ethical body and approval) for approval of protocol for using animal model in said experiments.
Comment 4: Authors to provide a description of the conditions under which the animals were kept, for example Temperature, humidity, the light dark cycle (including when lights were turned on). The temperature will be very important especially if it below 24°C. If the thermal environment is below the thermoneutral zone of the SD rats used. As such the rats will activate thermoregulatory effectors to increase heat gain thus spending their energy to maintain their body temperature. I am concerned that the associated increase in metabolic rate to enhance heat production for thermoregulation in environments below the thermoneutral zone have a bearing on the observed metabolic responses to the experimental treatments.
Comment 5: The methods for insulin tolerance testing and AICAR tolerance testing are not cleared, authors have to rewrite in the precise form.
RESULTS
Comment 6: Why did author use different number of mice for separate group, it should be equal in number to get the consistent results. Justify it?
Comment 7: Figures are not very clear; I would suggest to modify the format, so that readers can understand it easily.
DISCUSSION
Comment 8: How is this article more informative than the previously published ones? Justify it.
Comment 9: Authors have to include the limitation of their studies
Comment 10: Authors have to explore the future strategies of this studies
CONCLUSION
Comment 11: The authors should clarify the novelty of this article in the ‘Introduction’ and ‘Conclusion’ section.
Comment 12: Spelling check and grammatical error should be cross check.
Comment 13: Overall, the findings look promising, but the authors might need to conduct additional experimental studies to make their claims more scientific.
Author Response
We appreciate the constructive criticism of the reviewer, we have responded in a point-by-point fashion below, and believe the manuscript is improved as a result.
English Language Comment: We have reviewed the manuscript and corrected grammatical and typographical errors throughout.
Abstract – Comment #1: We have re-written the results section of the Abstract. We hope that the revisions describe our results with more clarity
Introduction – Comment #2: To more clearly communicate the novelty of our study we have added an opening sentence to paragraph two stating that no study has assessed the impact of PAR2 expression on age-related obesity.
Materials and Methods – Comment #3: We have added the Institutional Animal Care and Use Committee Protocol # and approval date to the revised manuscript
Materials and Methods – Comment #4: In the revised manuscript we have added information on the temperature and humidity of our animal housing room. Since our average temperature is below thermoneutrality, as is the case for many animal care facilities, we have added language at the beginning of the Discussion stating this is a limitation to our study. However, it is important to point out that all mice we exposed to identical environmental conditions throughout the study.
Materials and Methods – Comment #5: In the revised manuscript, we have added more details for the methods used for both the IT test and the AICAR test. We hope that these revisions make our methods clearer for the reader.
Results – Comment #6: In the revised manuscript, we have clarified the number of animals used in our study. Although the entire cohort of mice was used for the assessment of body composition, mice from each group (YG, AG, and PAR2KO-AG) were randomly selected for the GT test so that a total of 20 mice were used. These same 20 mice were also used for the IT and AT tests. Twenty mice are the maximal number of mice that we can study in one day and prevents any potential day-to-day variability that can occur with multiple experiments performed on different days. The AG and PAR2KO-AG mice that were not used for the GT, IT, and AT tests were used for a separate study. We should also note that these experiments were conducted in May 2020, at the very beginning of the COVID-19 pandemic so we were limited in the time and scope of studies that could be performed. Despite this short coming, we believe that we still have novel and interesting data to share with the scientific community.
Results – Comment #7: We have revised the figures so that they more clearly illustrate our data.
Discussion – Comment #8: In the revised manuscript, we have added language highlighting that our aging findings are quite different from previous published works using diet-induced obesity in young and PAR2KO mice.
Discussion – Comment #9: In the revised manuscript, we have added several sentences discussing the limitations to our study at the end of the first paragraph of the Discussion.
Discussion – Comment #10: In the Conclusions section of the revised manuscript, we have added language about future strategies of this study. Specifically, it would be interesting to know if PAR2 antagonism prevents age-related obesity.
Conclusion – Comment #11: In the Conclusions section of the revised manuscript, we have added language about the novelty of our study. We have also added similar language in the Introduction.
Conclusion – Comment #12: Both authors have reviewed the entire manuscript in hopes to catch all grammatical errors.
Conclusion – Comment #13: We appreciate that the reviewer’s comments about our findings being promising. Unfortunately, we do not have the resources to purchase additional PAR2KO mice and allow then to age for 18-20 months. We have acknowledged that further work is needed in this area.
Reviewer 2 Report
The author provided evidence supporting that genetic deletion of PAR2 did not ameliorate aging-induced adiposity in mice but improved glucose tolerance. However, the results suggest that aged mice in this study failed to develop glucose intolerance and insulin resistance, resulting in difficulties in assessing the impacts of PAR2 deletion on aging-induced insulin resistance.
Please change legends in line graphs, as they are quite similar to each other.
Please provide information about the diet used in this study
Please present a line graph that shows the changes in body weights throughout the study
Author Response
We appreciate the constructive criticism of the reviewer, we have responded in a point-by-point fashion below, and believe the manuscript is improved as a result.
In the revised manuscript, we have changed the line graphs so that our figure more clearly illustrate our data. Thank you.
We have added information regarding the diet to the Methods section of the revised manuscript, thank you for allowing us to clarify.
Unfortunately, we do not have serial measurements of body weight or body composition over time. We have added this limitation to the Discussion of the revised manuscript.
Round 2
Reviewer 1 Report
1. Materials and Methods:
- Line 70-71: is confusing: The mice were on a 12:12 hour light:dark cycle with the lights turning on at 0800 hours and off at 2000 hours. 2000 hours might not make any sense to the reader.
2. I would suggest changing words like novel/new in the manuscript with appropriate ones.
Author Response
We appreciate the reviewers efforts in further refining the manuscript. Our responses are provided below, with corresponding changes made to the manuscript.
1. Material and Methods: We agree that this statement is confusing. In the revised manuscript, we have changed the language to the lights turning on at 8:00 AM and turning off at 8:00 PM
2. In the revised manuscript, we have changed the wording from novel/new to “interesting”